# Different Mechanisms Cause Hypomethylation of Both *H19* and *KCNQ1OT1* Imprinted Differentially Methylated Regions in Two Cases of Silver–Russell Syndrome Spectrum

**DOI:** 10.3390/genes13101875

**Published:** 2022-10-16

**Authors:** Francesco Passaretti, Laura Pignata, Giuseppina Vitiello, Viola Alesi, Gemma D’Elia, Francesco Cecere, Fabio Acquaviva, Daniele De Brasi, Antonio Novelli, Andrea Riccio, Achille Iolascon, Flavia Cerrato

**Affiliations:** 1Department of Molecular Medicine and Medical Biotechnology, Università degli Studi di Napoli Federico II, 80131 Naples, Italy; 2U.O.C. Medical Genetics, Department of Translational Medical Sciences, A.O.U. Federico II, 80131 Naples, Italy; 3Department of Environmental Biological and Pharmaceutical Sciences and Technologies (DiSTABiF), Università degli Studi della Campania “Luigi Vanvitelli”, 81100 Caserta, Italy; 4Laboratory of Medical Genetics, Translational Cytogenomics Research Unit, Bambino Gesù Children Hospital, IRCCS, 00165 Rome, Italy; 5Institute of Genetics and Biophysics (IGB) “Adriano Buzzati-Traverso”, Consiglio Nazionale delle Ricerche (CNR), 80131 Naples, Italy; 6U.O.S. Medical Genetics, Pediatrics of Chronic and Multifactorial Diseases, A.O.R.N. Santobono-Pausilipon, 80129 Naples, Italy; 7CEINGE Advanced Biotechnology, 80131 Naples, Italy

**Keywords:** Silver–Russell syndrome, imprinting disorders, microduplication, differentially methylated regions, DNA methylation defects, 11p15.5 imprinted genes cluster

## Abstract

Silver–Russell syndrome is an imprinting disorder characterised by pre- and post-natal growth retardation and several heterogeneous molecular defects affecting different human genomic loci. In the majority of cases, the molecular defect is the loss of methylation (LOM) of the *H19/IGF2* differentially methylated region (DMR, also known as IC1) at the telomeric domain of the 11p15.5 imprinted genes cluster, which causes the altered expression of the growth controlling genes, *IGF2* and *H19*. Very rarely, the LOM also affects the *KCNQ1OT1* DMR (also known as IC2) at the centromeric domain, resulting in an SRS phenotype by an unknown mechanism. In this study, we report on two cases with SRS features and a LOM of either IC1 and IC2. In one case, this rare and complex epimutation was secondary to a de novo mosaic in cis maternal duplication, involving the entire telomeric 11p15.5 domain and part of the centromeric domain but lacking *CDKN1C*. In the second case, neither the no 11p15.5 copy number variant nor the maternal-effect subcortical maternal complex (SCMC) variant were found to be associated with the epimutation, suggesting that it arose as a primary event. Our findings further add to the complexity of the molecular genetics of SRS and indicate how the LOM in both 11p15.5 DMRs may result from different molecular mechanisms.

## 1. Introduction

Silver–Russell syndrome (SRS; OMIM #180860; also SRS2: #618905; SRS3: #616489; SRS4: #618907; SRS5: #618908. Estimated prevalence: 1:30,000–1:100,000) is a congenital disorder, characterised by intrauterine and post-natal growth retardation, relative macrocephaly at birth, feeding difficulties, a protruding forehead in early life, body asymmetry, and other less frequent features [1]. According to the Netchine-Harbison clinical score system (NH-CSS), clinical diagnosis is based on the presence of at least four out of the six most frequent features [1]. Recently, the definition of the Silver–Russell syndrome spectrum (SRSp) has been proposed to include all the cases with a clinical score < 4 but that still show clinical or molecular features of SRS [2].

The SRSp is caused by molecular changes affecting imprinted genes. The most frequently affected locus is at 11p15.5 and harbours two distinct imprinted domains overall extending for 1 Mb. A germline differentially methylated region (DMR) with the role of an imprinting centre (IC) regulating the monoallelic and parent-of-origin expression of the imprinted genes is present in each domain [3]. The telomeric domain encodes two genes with reciprocal imprinting: the insulin-like growth factor 2 (*IGF2*) gene is expressed from the paternal chromosome and encodes a foetal growth factor, and the *H19* gene is expressed from the maternal chromosome and encodes a non-coding RNA with growth inhibitory activity [4]. These two genes are under the control of the *H19-IGF2*:IG (intergenic)-DMR (also known as IC1) that is normally methylated on the paternal allele. The centromeric domain encodes the growth inhibitor *CDKN1C,* which is transcribed from the maternal chromosome, and its in cis repressor *KCNQ1OT1,* which is transcribed from the paternal chromosome and is a non-coding RNA [4]. The imprinting of the centromeric domain is controlled by the *KCNQ1OT1*:TSS (transcription start site)-DMR (also known as IC2) that overlaps the promoter of *KCNQ1OT1* and is normally methylated on the maternal allele.

The most frequent molecular defects associated with the SRSp include a number of genetic and epigenetic alterations of the 11p15 imprinted gene cluster (accounting for 30–60% of cases), overall causing an increased expression of the growth inhibitory genes *H19* and *CDKN1C*, and a decreased expression of the growth stimulatory gene *IGF2*. Maternal uniparental disomy of chromosome 7 (upd(7)mat) accounts for another 5–10% of cases [1]. Further alterations have been found at a lower frequency and affect either imprinted or non-imprinted genomic loci on different chromosomes (recently reviewed by Mackay and Temple [2]).

For the molecular diagnosis of the SRSp, it is recommended to test the DNA methylation of both IC1 and IC2 first, and in case of a positive result, determine if the epigenetic abnormality is associated with any CNV or UPD to estimate the recurrence risk [1]. Although IC1 loss of methylation (LOM) occurs most frequently as an isolated primary epimutation, it can be associated with IC2 gain of methylation (GOM) as consequence of upd(11)mat [5] or maternally inherited duplications of the entire cluster in rarer cases [6]. Isolated IC2 GOM is even rarer and can be associated with either maternally inherited duplications [7] or paternally inherited deletions [8]. In a few cases, IC1 LOM has been found to be associated with the LOM of additional imprinted DMRs [9]. This latter condition is known as multi-locus imprinting disturbances (MLID) and can be associated with the clinical manifestation of features not usually present in SRS and dependent on the affected loci [10]. In some of these cases, IC1 LOM is detected together with IC2 LOM, which is a hallmark of the overgrowth-associated Beckwith–Wiedemann syndrome spectrum (BWSp) and shows the phenotypic features of either the SRSp or BWSp [11]. In some families, MLID has been associated with loss-of-function or hypomorphic variants of maternal-effect genes encoding protein components of the subcortical maternal complex (SCMC) [12].

Here we report on two cases with SRSp features and both IC1 LOM and IC2 LOM. In one patient, the epimutations are associated with a mosaic de novo 1.9 Mb duplication of maternal origin, involving the entire telomeric 11p15.5 domain and part of the centromeric domain but lacking *CDKN1C*. In the second patient, no clearly pathogenic genetic change is associated with the epimutations, which thus appear to be of primary origin. Our findings further add to the complexity of the molecular genetics of the SRSp and indicate how the LOM of both 11p15.5 DMRs may result from different molecular mechanisms.

## 2. Material and Methods

### 2.1. Patients

Case 1. Proband 1 was born at term (37th w) from unrelated parents of Caucasian origin. The pregnancy was complicated by oligohydramnios. At birth, auxological parameters (weight 3.120 kg, 17th percentile/Z: −0.97; length 49 cm, 18th percentile/Z: −0.91; head circumference 35 cm, 53th percentile/Z: +0.08) were appropriate for gestational age. A spontaneously resolved mild atrial defect was reported at six months of age. A slight delay in language and acquisition of autonomous walking was observed in the first two years of life. Additionally, the face had a triangular shape with a prominent forehead. During childhood, the patient exhibited behavioural problems, relationship difficulties and learning difficulties. Since birth, the patient manifested gastrointestinal problems, including pyloric stenosis, intestinal occlusion, constipation, aversion to food and colic without any evidence of a precise organic cause. Between 5 and 8 years, height was <3rd percentile (Z: −2.8) and GH level was lower than normal (3.70 ng/mL after stimulation with arginine and 3.30 ng/mL after clonidine stimulation). At 8 years of age, the patient was treated with GH (Growth Hormone^®^) and during the first year of treatment he had already significantly gained height (32nd percentile/Z: −0.46). During the follow-up, a cystic formation of the pineal gland was demonstrated by brain MRI and a slight scoliosis was revealed by total spine X-ray. At our first clinical evaluation, the patient was 11 years and 11 months, his weight was 33.5 kg (13th percentile/Z: −1.13), height was 147.5 cm (44th percentile/Z: −0.15), cranial circumference was 54.3 cm (65th percentile/Z: +0.38), and BMI was 15.4 (11th percentile/Z: −1.20). We noted arched palate, slight prognathism, globose abdomen, several nevi on the back, clinodactyly of the 5th fingers, slight asymmetry of the lower limbs (<0.5 cm) and valgus hindfoot. Mild difficulties in motor coordination were also observed. A suspicion of SRS diagnosis was raised only in late childhood. However, according to the NH-CSS, only 2 of the 6 major criteria and some additional clinical features for SRS diagnosis were present. A molecular diagnosis of SRS with maternal 11p15 duplication was first indicated by the results of an SNP-array, which was proposed because of the neurodevelopmental delay.

Case 2. Proband 2, son of unrelated parents of Caucasian origin, was born at term (39th w) after a pregnancy complicated by intrauterine growth retardation (IUGR). At birth, growth deficiency was evident (weight 2.170 kg, <3rd percentile/Z: −3.19; length 46 cm, 1th percentile/Z: −2.31; head circumference 32 cm, 1th percentile/Z: −2.51). During the first year of life, he was breastfed with frequent episodes of regurgitation. Delay of the anterior fontanel closure and growth retardation were observed. The patient had normal psychomotor development and a very sociable character. Since the 8th month of life, heterometry of the lower limbs was evident. At our first clinical evaluation the patient aged 2 years and 2 months, weight was 9 kg (<3th percentile/Z: −3.15), height was 80.5 cm (1th percentile/Z: −2.26), cranial circumference was 46 cm (3th percentile/Z: −1.94), and BMI was 13.9 (3°/Z: −1.8). Dolichocephaly (17.5 cm of biparietal diameter, 19 cm of anteroposterior diameter), a face of triangular shape with protruding forehead, nasal hypoplasia, clinodactyly of the V fingers, flat philtrum, asymmetry of the lower limbs, and the right limb 1.5 cm longer than left were also evident. Clinical assessment according to the NH-CSS showed a rating of 4.

### 2.2. DNA Extraction

Peripheral blood lymphocyte (PBL) genomic DNA of probands and their parents was isolated by a QIAsymphony automatic extractor (QIAGEN, Hilden, Germany).

### 2.3. Methylation Analysis

Methylation-Specific Multiple Ligation-Dependent Probe Amplification (MS-MLPA) was performed on 50 ng of genomic PBL DNA by the commercially available assay, the SALSA MS-MLPA Probemix ME030-C3 or ME034-C1 (MRC-Holland, Amsterdam, The Netherlands), following manufacturer’s instructions. ABI 3500 Genetic Analyzer (Applied Biosystems, Foster City, CA, USA) was employed for the separation of the amplified products by capillary electrophoresis. Data were analysed using Coffalyser software (MRC-Holland, Amsterdam, The Netherlands).

Bisulfite conversion and Pyrosequencing analysis was carried out as previously reported [13]. Briefly, 1.5 μg of genomic DNA was treated with sodium bisulfite by the EpiTect Bisulfite kit (Qiagen, Hilden, Germany, cat. n. 59104) following the manufacturer’s protocol. PyroMark PCR kit (Qiagen, Hilden, Germany, cat. n. 978705) was used to amplify 200 ng of converted DNA. Fifteen μL of PCR product was used for the quantitative analysis of DNA methylation by pyrosequencing on a Pyromark Q48 Autoprep system with the PyroMark Q48 Adv. CpG Reagents (Qiagen, Hilden, Germany cat. n. 974022) and PyroMark Q48 Magnetic Beads (Qiagen, Hilden, Germany cat. n. 974203). Results were analysed by the Pyromark Q48 Autoprep software. The primer sequences have been previously reported [13].

Methylome array was performed on bisulphite converted PBL DNA of proband 2. Data were analysed using R version 4.1.0. β values were extracted from “idat” files by using the “champ.load” module of the “ChAMP” R package (v.2.22.0), with quality control options set as default and array type as “EPIC.” To adjust the β -values of type 2 probes, we applied BMIQ normalization with the default options for array type as “EPIC.” The coordinates of the imprinted DMRs were downloaded from http://www.humanimprints.net/ (accessed on 20 December 2021). Methylation profile was calculated as average of the methylation levels of their respective CpGs. Methylation levels of the patient were compared with 4 age-matched controls; a value deviating ± 3 standard deviation from the mean of the controls was considered as an aberrant methylation change. The raw and processed files are available on request.

### 2.4. SNP-Array

Single nucleotide polymorphism-array (SNP-array) analysis was performed on DNA of proband 1 and his parents using Infinium CytoSNP-850 K BeadChip (Illumina, San Diego, CA, USA) and in accordance with the manufacturer’s instructions. Array scanning data were generated by iScan system (Illumina, San Diego, CA, USA) and the results were analysed by Bluefuse Multi software (v 4.4).

### 2.5. FISH

Fluorescence in situ hybridization (FISH) analysis was performed to provide structural information on the microduplication. Locus-specific FISH analysis was performed on metaphases and nuclei obtained from PHA-stimulated lymphocytes, by means of a custom oligonucleotide probe (SureDesign, Agilent, Santa Clara, CA, USA), specifically designed within the duplicated region (11p15.5).

## 3. Results

Molecular testing for SRS was performed by MS-MLPA (ME030 assay) on the PBL DNA of the two patients who had received a clinical diagnosis of SRS (proband 2) or were suspected to be on the SRSp (proband 1; see Figure 1). The methylation analysis revealed the IC1 and IC2 LOM in both the probands, but it was less severe in proband 1 (methylation level 39.5% of both ICs) than in proband 2 (30%). The methylation defect was confirmed by a sodium bisulphite treatment and pyrosequencing in the probands and was excluded in their parents (Appendix A). The MLPA analysis also revealed a microduplication of 11p15.5, including both *H19* and *KCNQ1OT1* in proband 1. The last exons of *KCNQ1,* located downstream to IC2, as well as the *CDKN1C* gene were not included in the duplication. The copy number value (<1.5) suggested the presence of the duplication in the mosaic form. No CNV at 11p15.5 was detected in proband 2 (Figure 1).

The microduplication of proband 1 was further characterized by an SNP-array analysis, which confirmed the presence of a de novo 11p duplication in the mosaic form, with an extension of about 1.9 Mb, involving the entire telomeric SRS/BWS domain and only part of the centromeric domain (Figure 2A). The breakpoints were mapped at positions 795,147 and 2,712,286 of chr 11p (GRCh37). The lack of SNP-array probes within the IC2 region did not allow for the detection of IC2 CNVs. However, all four MS-MLPA probes for the IC2 CNV analysis revealed the duplication, demonstrating that at least two-thirds of the DMR and at least 200 bp centromeric to the transcription start site of *KCNQ1OT1* are included in the duplication. Furthermore, the analysis of the SNP genotypes of the duplicated region in the trio demonstrated that the duplication was of maternal origin (Figure 2B). The metaphase FISH analysis on the proband lymphoblasts demonstrated the hybridization of 11p15-specific probes only at the telomeric region of chromosome 11 and a signal of stronger intensity on one homologue in about half of the analysed cells, indicating that the duplication was in tandem and present in the mosaic form (Figure 2C).

As no CNV was detected at 11p15.5, further analyses were performed to investigate if the epigenetic defect was extended to further imprinted loci in proband 2. The multi-locus MS-MLPA (ME034) did not reveal any epigenetic defects or CNV at additional disease-associated imprinted loci (Appendix A). The methylation analysis of 39 imprinted DMRs by an Illumina Infinium EPIC methylation array confirmed the 11p15.5 LOM and demonstrated a further slight LOM at *NAP1L5*:TSS-DMR on chr 4q22 and a slight GOM affecting the *INPP5F*:Int2-DMR on chr 10q26 (Appendix A). Overall, the MS-MLPA and methylome results showed that the 11p15.5 ICs epimutations of proband 2 are associated with an MLID profile. Whole-exome sequencing was performed on the DNA of proband 2’s mother, but no clearly pathogenic variant in the SCMC genes was identified. 

## 4. Discussion

Among the known imprinting disorders, SRS is probably the one with the most heterogeneous molecular genetics, a fact that makes molecular diagnosis very challenging. Accordingly, five different OMIM ID entries have been associated with SRS so far (https://omim.org (accessed on 2 March 2022); chromosome 11p15.5 (IC1: SRS1, #180,860; IGF2: SRS3, #616,489), 7p13-q32 (SRS2, #618,905), 8q12.1 (SRS4, PLAG1, #618,907), and 12q14 (SRS5, HMGA2, #618,908)). The majority of the cases belong to the first subgroup which is further characterized by different molecular mechanisms underlying IC1 LOM. In this study, we report two cases that further add to the molecular complexity of SRS1. Both patients showed a LOM of either IC1 and IC2, but this complex epimutation was associated with a de novo mosaic in cis maternal duplication in the former case, but no 11p15.5 CNV or maternal-effect SCMC variant in the latter case.

Case 1. Proband 1 represents a peculiar case both due to his clinical features and molecular defect. Concerning the clinical diagnosis, the proband does not fulfil the clinical criteria of SRS according to NH-CSS, as only two (post-natal growth retardation and protruding forehead) out of six main features are reported. Although molecular testing demonstrated IC1 LOM, this case does not show the typical defects of SRS, because the maternal duplication is not extended to the whole centromeric 11p15.5 domain and particularly does not include the *CDKN1C* gene. Nevertheless, several additional features of SRS (slight asymmetry, fifth finger clinodactyly, triangular face, scoliosis, speech and motor delay) are present, and the duplication includes the telomeric domain with the *H19* gene. According to a definition recently proposed by Mackay and Temple, this case might be possibly classified within the Silver–Russell syndrome spectrum (SRSp) [2].

Although the duplication has a maternal origin, IC2 is affected by the LOM instead of the expected GOM due to its maternal methylation. This discrepancy could result from a position effect of nearby loci on the duplicated IC2, which is located at the end of the duplicated region (Figure 3). In three previously reported cases, partial duplications of the centromeric domain were associated with IC2 LOM but with the BWS phenotype when maternally inherited [14,15]. Indeed, IC2 LOM is expected to cause the activation of *KCNQ1OT1*, the repression of *CDKN1C,* and most likely, overgrowth. Our patient shows normal growth parameters at birth and a mild SRS phenotype in infancy, which may possibly result from a compensatory effect between the growth stimulation deriving from IC2 LOM and the growth inhibition caused by *H19* duplication (Figure 3). Alternatively, it is possible that the duplicated *KCNQ1OT1* is not expressed because it lacks part of its promoter or because *KCNQ1OT1* is too far away (>2–4 Mb) from *CDKN1C* to exert its repressive action in cis because it is inserted in the telomeric breakpoint.

Maternal 11p15.5 duplications associated with SRS are generally germline and affect the entire imprinted cluster or are restricted to its centromeric domain [1,7,14], supporting a role of *CDKN1C* in SRSp pathogenesis. Although *H19*’s function as growth inhibitor is indicated by several mouse studies [16,17], the role of the telomeric domain in SRS is less clear. A case of the maternal duplication of the entire telomeric domain was associated with a normal phenotype [18]. Three further cases with the partial duplication of the telomeric domain, involving *H19* but not *IGF2*, displayed growth retardation, but two of them had additional cytogenetic anomalies that might also explain this phenotype [14,15,19]. Finally, a somatic maternal 11p15 duplication has been identified on the smaller side of the face of a patient with body asymmetry [20]. The duplication of our patient is mosaic, maternal, includes the entire telomeric domain and the centromeric domain but not *CDKN1C*, and is not associated with any further cytogenetic anomalies. This finding supports a role of the duplicated *H19* in SRSp pathogenesis, although the involvement of further genes located in the duplicated region cannot be completely ruled out.

By querying the Decipher database, we have found that about fifty genes in addition to the 11p15.5 imprinted gene cluster are included in the duplication of proband 1 (https://www.deciphergenomics.org/search/patients/results?q=grch37%3A11%3A795147-1941891, accessed on 15 September 2022). Twenty-six cases are reported to be carriers of duplications overlapping this region. Of the very heterogeneous clinical features of these individuals, a few are present in proband 1, such as mild developmental delay, short stature, and clinodactyly. However, in most of these cases, the duplication also involves the imprinted gene cluster, making a correct genotype–phenotype correlation difficult. On the other hand, it cannot be excluded that genes outside the 11p15.5 imprinted gene cluster may contribute to the atypical clinical phenotype of proband 1.

Case 2. IC1 LOM and IC2 LOM are hallmarks of SRS and BWS, respectively. Nevertheless, both these epimutations are associated with severe growth retardation in proband 2. IC1 LOM+IC2 LOM has previously been reported in several cases, whose phenotype was either SRS or BWS (Table 1). As in proband 2, the epigenetic defect is always partial, supporting the hypothesis of errors in imprinting maintenance arising post-zygotically. The resulting epigenetic and gene expression mosaicism probably explain the divergent clinical features as well as the frequent body asymmetry of the affected individuals [11,12]. Most of the IC1 LOM+IC2 LOM patients show the hypomethylation of additional DMRs and have been classified as MLID cases (Table 1). Although further studies are needed to clarify genotype–epigenotype correlations, the current idea to explain the clinical outcome of the MLID patients is the epidominance hypothesis, which is based on the mosaic form of the multiple methylation changes in BWS and SRS [11]. According to this hypothesis, the main clinical presentation of the patient is caused by the imprinted locus that is mostly affected, while the other affected loci may possibly contribute to atypical features. For example, the LOM of the GNAS locus has been found associated to 11p15.5 IC2 LOM in MLID patients with BWS and pseudohypoparathyroidism 1B [21] or hypocalcemia [22]. In SRS-MLID, the most affected DMRs other than IC1 are *MEST*:alt-TSS-DMR and *GRB10*:alt-TSS-DMR [12] (Table 1), although we did not find abnormal methylation at these loci in our patient. Some cases of BWS and SRS with MLID have been associated with maternal variants of the SCMC genes [9,12,22,23,24,25,26,27], but the whole-exome sequencing did not identify any such variant in our case.

## Figures and Tables

**Figure 1 genes-13-01875-f001:**
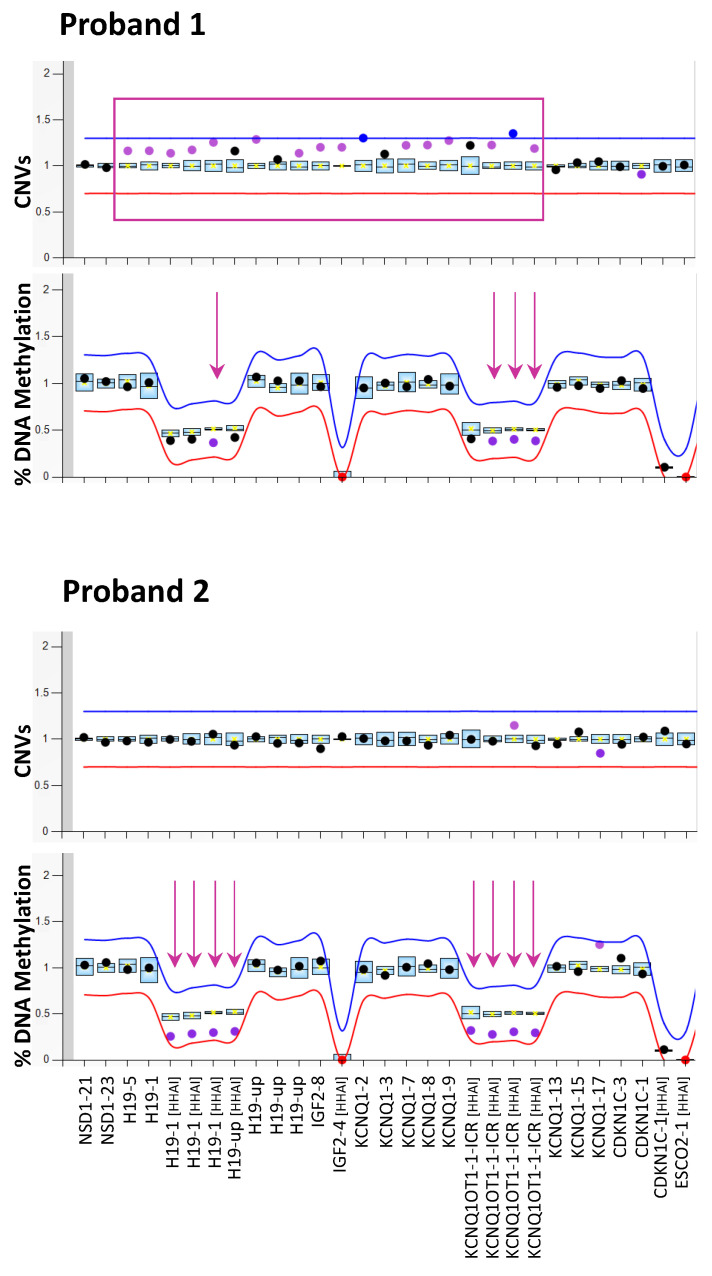
Molecular testing of SRS by MS-MLPA. Copy number (CNVs, upper panels) and DNA methylation (lower panels) of the 11p15 imprinting genes cluster analysed on PBL DNA of both the probands by ME030 BWS/SRS diagnostic kit. The mean values of three control subjects were used for assessment of relative copy number and methylation percentage. Arrows indicate the probes detecting the methylation defect; rectangles indicate the probes detecting the duplication.

**Figure 2 genes-13-01875-f002:**
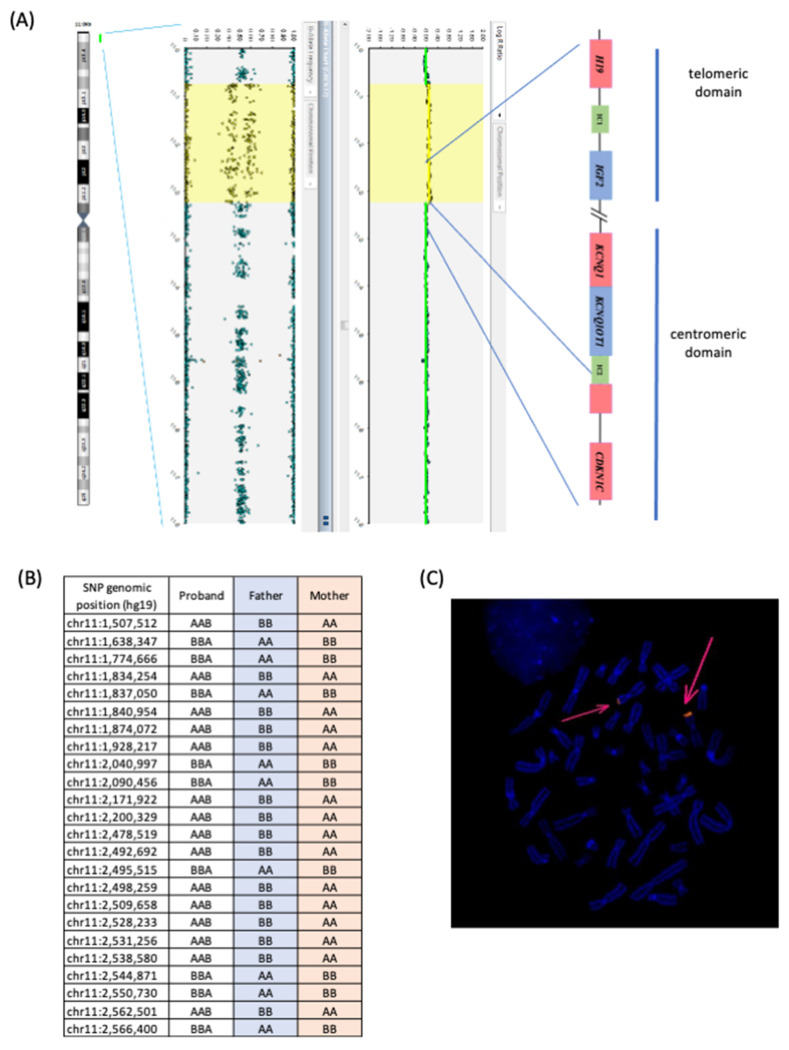
Genetic characterization of the 11p15 duplication in proband 1. (**A**) SNP-array analysis showing a mosaic duplication (in yellow) extending from 795,147 to 2,712,286 of chromosome 11 (rsa[GRCh37] 11p15.5(795,147_2,712,286)×3 [0.5]). (**B**) Informative SNPs of the TRIO mapping in the duplicated region revealed the maternal origin of the duplication. (**C**) FISH analysis on methaphases and nuclei showing the probe signal mapping only at chromosomes 11 (pink arrows). The signal was detectable in about 50% of the cells.

**Figure 3 genes-13-01875-f003:**
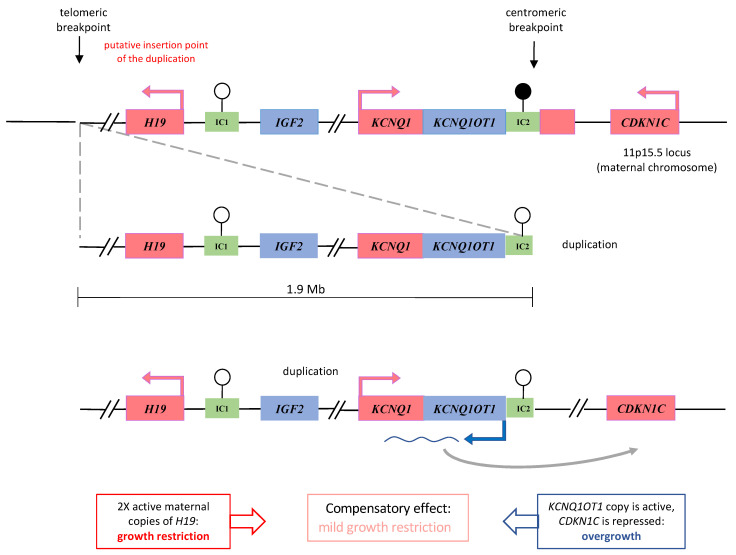
Diagram illustrating a hypothesis to explain the phenotype of the proband 1: a compensatory effect between growth stimulation deriving from IC2 LOM and growth inhibition caused by *H19* duplication. The duplicated region is depicted separately from the chromosome and connected to the breakpoints by dashed lines. Black arrows indicate the genomic position of the breakpoints. Thicker dashed line represents the most probable location of the duplication to better explain the IC2 LOM. Rectangles represent maternally- (in red) and paternally- (in blue) expressed imprinted genes and imprinting centres (IC, in green). Black lollipops indicate methylated IC; white lollipops, unmethylated IC.

**Table 1 genes-13-01875-t001:** Epigenotype and clinical features of patients with the LOM in both IC1 and IC2 reported in literature.

ID	Clinical Features	Hypo/Hyper Methylated Loci	References
SRS	SGA, PNGR, protruding forehead, body asymmetry, dolichocephaly, triangular face, nasal hypoplasia, clinodactyly of the V fingers, flat philtrum	*H19, KCNQ1OT1, NAP1L5, INPP5F*	Proband 2 of present study
SRS	SGA, PNGR, relative macrocephaly at birth, protruding forehead, body asymmetry, feeding difficulties	*H19, KCNQ1OT1, MEST*	[11,12]
SRS	Birth weight at 27 wg 465 g, OFC 32 cm. PNGR, respiratory support for 2 months, gastric tube feeding for first year. Microcephaly, precocious puberty, dysmorphism. Developmental delay. 47, XXY	*H19, KCNQ1OT1, GRB10, MEST, MEG3, GNAS-AS, GNAS*	[11,12]
SRS	Discordant monozygotic twin, kidney failure in infancy, bilateral renal dysplasia	*H19, KCNQ1OT1, PLAGL1, IGF2R, IGF1R, PEG3, GNAS-AS*	[11,12]
SRS	PNGR, relative macrocephaly, facial gestalt (prominent forehead, triangular face, downturned corners of the mouth, micrognathia), asymmetry and clinodactyly of the fifth digit	*H19, KCNQ1OT1*	[11,12]
SRS	N/A	*H19, KCNQ1OT1, MEG3*	[11,12]
SRS	N/A	*H19, KCNQ1OT1, MEG3, IGF2R*	[11,12]
SRS	N/A	*H19, KCNQ1OT1, PLAGL1, MEST, MEG3*	[11,12]
BWS	Polyhydramnios, macroglossia, umbilical ernia, hypoglicaemia, naevus flammeus	*H19, KCNQ1OT1, MEG3, GNAS-AS, GNAS, PEG3, PLAGL1 GRB10, MEST*	[11,12]
BWS	Macroglossia, umbilical ernia, hypoglicaemia hemihypertrophy, naevus flammeus, ear creases, low birth weight, hypocalcemia, facial dysmorphism, slight cognitive delay	*GRB10, MEST, H19, KCNQ1OT1, MEG3, GNAS-AS, GNAS, PEG3, PLAGL1, SNRPN*	[11,12]
BWS	Macroglossia, cheek and tongue right-side hemihyperplasia, naevus flammeus, diastasis recti	*H19, KCNQ1OT1, PLAGL1, MEST, GNAS-AS, GNAS*	[11,12]
BWS	BW 90th–97th centile, macrosomia, macroglossia, asymmetry, naevus flammeus, ear creases, developmental delay	*H19, KCNQ1OT1, GRB10, MEST, IGF2R, IGF1R*	[11,12]
BWS	N/A	*H19, KCNQ1OT1, MEG3, GNAS-AS, GNAS*	[11,12]
BWS	N/A	*H19, KCNQ1OT1, MEG3, SNRPN, PEG3*	[11,12]
BWS	N/A	*H19, KCNQ1OT1, PLAGL1, MEST, GNAS-AS, GNAS*	[11,12]
BWS	N/A	*H19, KCNQ1OT1, PLAGL1, MEST, IGF2R*	[11,12]

SGA: small for gestational age; PNGR: postnatal growth retardation; BW: birth weight; N/A: not available. PLAGL1, PLAGL1:alt-TSS-DMR; GRB10, GRB10:alt-TSS-DMR; MEST, MEST:alt-TSS-DMR; H19, H19/IGF2:IG-DMR; KCNQ1OT1, KCNQ1OT1:TSS-DMR; MEG3, MEG3/DLK1:IG-DMR;NAP1L5, NAP1L5:TSS-DMR; INPP5F, INPP5:Int2-DMR; GNAS-AS, GNAS-AS1-TSS-DMR; GNAS, GNAS A/B:TSS-DMR; SNRPN, SNURF:TSS-DMR; PEG3, *PEG3*:TSS-DMR.

## Data Availability

The data that support the findings of this study are available on request from the corresponding author.

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
