# Peer review of "Different Mechanisms Cause Hypomethylation of Both H19 and KCNQ1OT1 Imprinted Differentially Methylated Regions in Two Cases of Silver–Russell Syndrome Spectrum"

_genes, 2022, doi:10.3390/genes13101875_

Round 1

Reviewer 1 Report

This is an interesting topic and well written manuscript.

However I have some concerns about the results :

It seems that there is a discrepancy between the coordinates of the duplicated fragment in the main text (11 :795,147-2,712,286) and figures 2 and 3. Indeed, it seems that the duplicated fragment includes only a part of KCNQ1OT1 and not ICR2 which is located in the KCNQ1OT1 promotor (hg19 : chr11:2,719,948-2,722,259 according to Monk et al, PMID 27911167). Furthermore, in figure 2B, I was wandering if informative SNPs after position 2,200,329 are present (as the duplicated fragment extends to 2,712,286 according to the main text). 

In the discussion part, no information is provided about the previously reported partial duplication of the centromeric domain. Indeed, such duplications (see Heide et al 2018, Demars et al 2011) are associated with IC2 loss of methylation and a phenotype of Beckwith Wiedemann syndrome when maternally inherited. This could explain why birth parameters could be in the normal range.

Author Response

We wish to thank the referees for their valuable comments that helped us to improve the quality of our manuscript. Please, find below our responses to the comments.

This is an interesting topic and well written manuscript.

However, I have some concerns about the results:

It seems that there is a discrepancy between the coordinates of the duplicated fragment in the main text (11 :795,147-2,712,286) and figures 2 and 3. Indeed, it seems that the duplicated fragment includes only a part of KCNQ1OT1 and not ICR2 which is located in the KCNQ1OT1 promotor (hg19 : chr11:2,719,948-2,722,259 according to Monk et al, PMID 27911167). Furthermore, in figure 2B, I was wandering if informative SNPs after position 2,200,329 are present (as the duplicated fragment extends to 2,712,286 according to the main text). 

R#1-1. We added further informative SNPs to the list of fig 2B, but unfortunately the platform used for the SNP-array (Infinium CytoSNP‐850 K BeadChip (Illumina, San Diego, CA)) does not contain probes hybridizing to the ICR2 region. However, all the four probes of the SALSA MS-MLPA Probemix ME030-C3 for CNVs detection of ICR2 reveal the duplication (fig. 1). These probes cover the two-thirds of the DMR. Accordingly, we added the following sentences in Results:

“The breakpoints mapped at positions 795,147 and 2,712,286 of chr 11p (GRCh37). The lack of SNP-array probes  within the IC2 region does not allow detection of IC2 CNVs. However, all four MS-MLPA probes for IC2 CNV analysis revealed the duplication, demonstrating that at least two-thirds of the DMR and at least 200 bp centromeric to the transcription start site of KCNQ1OT1 are included in the duplication. Furthermore, …”.

We also modified the figures 2 and 3 by moving the centromeric breakpoint of the duplication inside the IC2, according to the MS-MLPA CNV detection results.

In the discussion part, no information is provided about the previously reported partial duplication of the centromeric domain. Indeed, such duplications (see Heide et al 2018, Demars et al 2011) are associated with IC2 loss of methylation and a phenotype of Beckwith Wiedemann syndrome when maternally inherited. This could explain why birth parameters could be in the normal range.

R#1-2. We have discussed the cases 18 and 19 reported by Heide et al (2018) and the case L65P reported by Demars et al (2011) as follows:

“Although the duplication has maternal origin, IC2 is affected by LOM instead of the expected GOM due to its maternal methylation. This discrepancy could result from a position effect of nearby loci on the duplicated IC2, that is located at the end of the duplicated region (Figure 3). In three previously reported cases, partial duplications of the centromeric domain were associated with IC2 LOM but BWS phenotype when maternally inherited [14,15]. Indeed, IC2 LOM is expected to cause activation of KCNQ1OT1 and repression of CDKN1C and likely overgrowth. Our patient shows normal growth parameters at birth and mild SRS phenotype in infancy, which may possibly result from a compensatory effect between growth stimulation deriving from IC2 LOM and growth inhibition caused by H19 duplication. Alternatively, it is possible that the duplicated KCNQ1OT1 is not expressed because lacking part of its promoter or that KCNQ1OT1 is too far away (>2-4 Mb) from CDKN1C to exert its repressive action in cisbecause inserted in the telomeric breakpoint.

Also, we have modified the figure 3 by adding a scheme illustrating the hypothesis of the compensatory effect between the IC2 LOM/CDKN1C repression and the two active copies of H19.

Reviewer 2 Report

In their paper, Passaretti et al. describe two patients with the rare finding of a loss of methylation of both differentially methylated regions in 11p15.5 (IC1 and IC2 LOM) and put these findings in the context of the literature.

In principle, the paper is interesting and, in my view, carefully edited, but because ultimately no new findings are presented, more information would have to be additionally incorporated into the text for acceptance as an article: This would provide an additional value to the readership:

-       patient 1 cannot receive a clinical diagnosis of SRS because, as the authors rightly say, the Netchine Harbison Score only gives 2. This should be made appropriately clearer in the text e.g. first sentence of the results.

-       the clinical description of patient 1 also differs significantly from Silver-Russell patients. Both the physical measurements, especially at birth, and the syndromal developmental disorder exhibited by the patient are very atypical for SRS. In this context genes located in the duplication already associated with syndromal disorders away from SRS should be discussed more deeply.

-       - likewise, a figure comparing the patient to those patients already known in the literature with microduplications of the region could help (Decipher database).

-       patient 2 shows evidence of association with MLID in the epic array. Even if no maternal effect mutation could be detected, this is an important finding and should be presented more clearly, e.g. discussion first paragraph last sentence: no evident genetic abnormality is not correct.

-       also missing here is a review of already known other patients with LOM in both IC1 and IC2 and the indication that these were usually MLID patients. Perhaps the authors could not only provide the citations but also give a more detailed overview at this point (table?).

Author Response

We wish to thank the referees for their valuable comments that helped us to improve the quality of our manuscript. Please, find below our responses to the comments.

In their paper, Passaretti et al. describe two patients with the rare finding of a loss of methylation of both differentially methylated regions in 11p15.5 (IC1 and IC2 LOM) and put these findings in the context of the literature. 

In principle, the paper is interesting and, in my view, carefully edited, but because ultimately no new findings are presented, more information would have to be additionally incorporated into the text for acceptance as an article: This would provide an additional value to the readership: 

-  patient 1 cannot receive a clinical diagnosis of SRS because, as the authors rightly say, the Netchine Harbison Score only gives 2. This should be made appropriately clearer in the text e.g. first sentence of the results. 

R#2-1. We modified the first sentence of the results as follows:

“Molecular testing for SRS was performed by MS-MLPA (ME030 assay) on PBL DNA of the two patients who had received a clinical diagnosis of SRS (proband 2) or suspicion of SRSp (proband 1; see Figure 1)”.

-   the clinical description of patient 1 also differs significantly from Silver-Russell patients. Both the physical measurements, especially at birth, and the syndromal developmental disorder exhibited by the patient are very atypical for SRS. In this context genes located in the duplication already associated with syndromal disorders away from SRS should be discussed more deeply. 

-   likewise, a figure comparing the patient to those patients already known in the literature with microduplications of the region could help (Decipher database). 

R#2-2. We addressed these points adding the following paragraph in the Discussion. For a better and more complete description of the cases with duplications overlapping the region under study, we refer to the Decipher database (https://www.deciphergenomics.org/).

“By querying the Decipher database, we have found that about fifty genes in addition to the 11p15.5 imprinted gene cluster are included in the duplication of the proband 1 (https://www.deciphergenomics.org/search/patients/results?q=grch37%3A11%3A795147-1941891). Twenty-six cases are reported to be carriers of duplications overlapping this region. Of the very heterogeneous clinical features of these individuals, a few, such as mild developmental delay, short stature, clinodactyly, are present in proband 1. However, in most of these cases the duplication also involves the imprinted gene cluster, making difficult a correct genotype-phenotype correlation. On the other hand, it cannot be excluded that genes outside the 11p15.5 imprinted gene cluster may contribute to the atypical clinical phenotype of the proband 1.”.

-    patient 2 shows evidence of association with MLID in the epic array. Even if no maternal effect mutation could be detected, this is an important finding and should be presented more clearly, e.g. discussion first paragraph last sentence: no evident genetic abnormality is not correct. 

R#2-3. We changed the sentence “no evident genetic abnormality” in the Discussion and Abstract as follow:

Discussion: “Both patients showed LOM of either IC1 and IC2, but this complex epimutation was associated with a de novo mosaic in cis maternal duplication in the former case, but no 11p15.5 CNV or maternal-effect SCMC variant in the latter case.”.

Abstract: “In one case, this rare and complex epimutation was secondary to a de novo mosaic in cis maternal duplication involving the entire telomeric 11p15.5 domain and part of the centromeric domain but lacking CDKN1C. In the second case, no 11p15.5 copy number variant nor maternal-effect subcortical maternal complex (SCMC) variant was found associated with the epimutation suggesting that it arised as primary event.”.

-    also missing here is a review of already known other patients with LOM in both IC1 and IC2 and the indication that these were usually MLID patients. Perhaps the authors could not only provide the citations but also give a more detailed overview at this point (table?).

R#2-4. We added a table (Table 1) listing the epigenotype and clinical features all the MLID patients with IC1 and IC2 LOM reported in literature so far. Also, we modified the paragraph “Case 2” of Discussion as follows:

“Case 2. IC1 LOM and IC2 LOM are hallmarks of SRS and BWS, respectively. Nevertheless, both these epimutations are associated with severe growth retardation in proband 2. IC1 LOM+IC2 LOM has previously been reported in several cases, whose clinical phenotype was either SRS or BWS (Table 1) [11,12]. As in proband 2, the epigenetic defect is always partial supporting the hypothesis of errors in imprinting maintenance arising post-zygotically. The resulting epigenetic and gene expression mosaicism probably explain the divergent clinical features as well as the frequent body asymmetry of the affected individuals[11,12]. Most of the IC1 LOM+IC2 LOM patients show hypomethylation of additional DMRs and have been classified as MLID cases (Table 1). The most frequently hypomethylated DMRs are GNAS DMRs and PLAGL1:alt-TSS-DMR in BWS, and MEST:alt-TSS-DMR and GRB10:alt-TSS-DMR  in SRS (Table 1) [12], although we did not find abnormal methylation at these loci in our patient. Some cases of BWS and SRS with MLID have been associated with maternal variants of the SCMC genes[9,12,21-25], but whole-exome sequencing did not identify any such variant in our case. “.

Reviewer 3 Report

Please provide figure 1, 2 and 3. Without this manuscript in unaceptable for publication.

Author Response

We wish to thank the referees for their valuable comments that helped us to improve the quality of our manuscript. Please, find below our responses to the comments.

Please provide figure 1, 2 and 3. Without this manuscript in unaceptable for publication.

R#3-1. The figures are now reported in the text of the manuscript. Very sorry for this inconvinience

Reviewer 4 Report

Passaretti and colleagues present the complexity of the molecular genetics of Silver-Russell syndrome. The study is of value to the literature. However, this manuscript needs a significant degree of editing (see below).

Abstract

1. “Our findings further add to the complexity of the molecular genetics of SRS and indicate how LOM of both 11p15.5 DMRs may result from different molecular mechanisms.”

=> DMR should be presented as differentially methylated regions (DMRs).

Introduction

1.” For the molecular diagnosis of SRSp it is recommended to test DNA methylation of both IC1 and IC2 first, and in case of a positive result, determine if the epigenetic abnor-mality is associated with any CNV or UPD to estimate recurrence risk1.”

=> risk1 should be risk [1].

Material and Methods

1. “Case 1. Proband 1 was born at term (37°w) from unrelated parents of Caucasian origin.”

=> 37°w should be 37thw.

2. Are there any patients’ photos in this manuscript?

Figures

1. There are no figure 1, figure 2 and figure 3 in this manuscript.

Discussion

1. “Some cases of BWS and SRS with MLID have been as-sociated with maternal variants of the SCMC genes.”

=> Are there some references about these cases?

2. “This could be explained by the insertion of the duplication in the telomeric breakpoint that is probably too far (>2-4 Mb) from KCNQ1OT1 to exert its repressive action on CDKN1C (Figure 3).”

=> Are there some references to support your view that the duplicated KCNQ1OT1 does not repress CDKN1C on the maternal chromosome?

Author Response

We wish to thank the referees for their valuable comments that helped us to improve the quality of our manuscript. Please, find below our responses to the comments.

Passaretti and colleagues present the complexity of the molecular genetics of Silver-Russell syndrome. The study is of value to the literature. However, this manuscript needs a significant degree of editing (see below).

Abstract

  1. “Our findings further add to the complexity of the molecular genetics of SRS and indicate how LOM of both 11p15.5 DMRs may result from different molecular mechanisms.”

=> DMR should be presented as differentially methylated regions (DMRs).

R#4-1. We modified the second sentence of the abstract as follows: “In the majority of cases the defect is the loss of methylation (LOM) of the H19/IGF2 differentially methylated region (DMR, also known as IC1…”

Introduction

1.” For the molecular diagnosis of SRSp it is recommended to test DNA methylation of both IC1 and IC2 first, and in case of a positive result, determine if the epigenetic abnormality is associated with any CNV or UPD to estimate recurrence risk1.”

=> risk1 should be risk [1].

R#4-2. The reference number was corrected by replacing risk1 with risk [1].

Material and Methods

  1. “Case 1. Proband 1 was born at term (37°w) from unrelated parents of Caucasian origin.”

=> 37°w should be 37thw.

R#4-3. 37°w was replaced with 37thw

  1. Are there any patients’ photos in this manuscript?

R#4-4. No, there are no patients’s photos reported in the manuscript.

Figures

  1. There are no figure 1, figure 2 and figure 3 in this manuscript.

R#4-5. The figures are now reported in the text of the manuscript. Very sorry for this inconvinience.

Discussion

  1. “Some cases of BWS and SRS with MLID have been as-sociated with maternal variants of the SCMC genes.”

=> Are there some references about these cases?

R#4-6. The following references were added to this sentence:

Begemann, M.; Rezwan, F.I.; Beygo, J.; Docherty, L.E.; Kolarova, J.; Schroeder, C.; Buiting, K.; Chokkalingam, K.; Degenhardt, F.; Wakeling, E.L.; et al. Maternal variants in. J Med Genet 2018, 55, 497-504, doi:10.1136/jmedgenet-2017-105190.

Pignata, L.; Cecere, F.; Verma, A.; Hay Mele, B.; Monticelli, M.; Acurzio, B.; Giaccari, C.; Sparago, A.; Hernandez Mora, J.R.; Monteagudo-Sánchez, A.; et al. Novel genetic variants of KHDC3L and other members of the subcortical maternal complex associated with Beckwith-Wiedemann syndrome or Pseudohypoparathyroidism 1B and multi-locus imprinting disturbances. Clin Epigenetics 2022, 14, 71, doi:10.1186/s13148-022-01292-w.

 Eggermann, T.; Yapici, E.; Bliek, J.; Pereda, A.; Begemann, M.; Russo, S.; Tannorella, P.; Calzari, L.; de Nanclares, G.P.; Lombardi, P.; et al. Trans-acting genetic variants causing multilocus imprinting disturbance (MLID): common mechanisms and consequences. Clin Epigenetics 2022, 14, 41, doi:10.1186/s13148-022-01259-x.

Docherty, L.E.; Rezwan, F.I.; Poole, R.L.; Turner, C.L.; Kivuva, E.; Maher, E.R.; Smithson, S.F.; Hamilton-Shield, J.P.; Patalan, M.; Gizewska, M.; et al. Mutations in NLRP5 are associated with reproductive wastage and multilocus imprinting disorders in humans. Nat Commun 2015, 6, 8086, doi:10.1038/ncomms9086.

Sparago, A.; Verma, A.; Patricelli, M.G.; Pignata, L.; Russo, S.; Calzari, L.; De Francesco, N.; Del Prete, R.; Palumbo, O.; Carella, M.; et al. The phenotypic variations of multi-locus imprinting disturbances associated with maternal-effect variants of NLRP5 range from overt imprinting disorder to apparently healthy phenotype. Clin Epigenetics 2019, 11, 190, doi:10.1186/s13148-019-0760-8.

Cubellis, M.V.; Pignata, L.; Verma, A.; Sparago, A.; Del Prete, R.; Monticelli, M.; Calzari, L.; Antona, V.; Melis, D.; Tenconi, R.; et al. Loss-of-function maternal-effect mutations of PADI6 are associated with familial and sporadic Beckwith-Wiedemann syndrome with multi-locus imprinting disturbance. Clin Epigenetics 2020, 12, 139, doi:10.1186/s13148-020-00925-2.

Eggermann, T.; Kadgien, G.; Begemann, M.; Elbracht, M. Biallelic PADI6 variants cause multilocus imprinting disturbances and miscarriages in the same family. Eur J Hum Genet 2021, 29, 575-580, doi:10.1038/s41431-020-00762-0.

Tannorella, P.; Calzari, L.; Daolio, C.; Mainini, E.; Vimercati, A.; Gentilini, D.; Soli, F.; Pedrolli, A.; Bonati, M.T.; Larizza, L.; et al. Germline variants in genes of the subcortical maternal complex and Multilocus Imprinting Disturbance are associated with miscarriage/infertility or Beckwith-Wiedemann progeny. Clin Epigenetics 2022, 14, 43, doi:10.1186/s13148-022-01262-2.

  1. “This could be explained by the insertion of the duplication in the telomeric breakpoint that is probably too far (>2-4 Mb) from KCNQ1OT1 to exert its repressive action on CDKN1C (Figure 3).”

=> Are there some references to support your view that the duplicated KCNQ1OT1 does not repress CDKN1C on the maternal chromosome?

R#4-7. This point is discussed above. Please, see answer R#1-2 to Reviewer 1.

Round 2

Reviewer 3 Report

Presented case reports include practical use of the results of methylation variation in a range of loci combined with molecular cytogenetics and SNP microarray analysis in order to resolve the complexity of epigenetic background of disorders with disturbed mechanisms of imprinting. The use of aforementioned approaches allowed to reveal new molecular signs of SRS syndrome in form of de novo mosaic in cis maternal duplication including telomeric and centromeric domains without CDKN1C locus. I found this epigenetic studies as complete.

Author Response

We thank the reviewer for the nice comments on our manuscript.

Reviewer 4 Report

Bertini and colleagues present DLG2 implications in neuropsychiatric disorders The study is of value to the literature. However, this manuscript needs a minor degree of editing (see below).

Discussion

"Most of the IC1 LOM+IC2 LOM patients show hypomethylation of additional DMRs and have been classified as MLID cases (Table 1). The most frequently hypomethylated DMRs are GNAS DMRs and PLAGL1:alt-TSS-DMR in BWS, and MEST:alt-TSS-DMR and GRB10:alt-TSS-DMR in SRS[12] (Table 1), although we did not find abnormal methylation at these loci in our patient."

=> Could we explain that why different DMRs could make various symptoms in SRS and BWS patients?

Author Response

Although further studies are needed to clarify genotype-epigenotype correlations, the current idea to explain the clinical outcome of the MLID patients is the epidominance hypothesis that is based on the mosaic form of the multiple methylation changes in BWS and SRS (Azzi et al., 2009). According to this hypothesis, the main clinical presentation of the patient is caused by the imprinted locus that is mostly affected, while the other affected loci may possibly contribute to atypical features. For example, LOM of GNAS locus has been found associated to 11p15.5 IC2 LOM in MLID patients with BWS and pseudohypoparathyroidism 1B (Sano et al., 2016) or hypocalcemia (Sparago et al., 2019). Furthermore, LOM at 14q32 and 11p15.5 has been found in a patient affected by BWS and Temple syndrome (Grosvenor et al., 2021). Finally, BWS patients with IC1 LOM+IC2 LOM are often not affected by macrosomia or they even show low birth weight (table 1) possibly because of a compensatory effect between growth inhibition due to IC1 LOM and growth stimulation due to IC2 LOM.

In order to better explain this concept, we modified the paragraph “Case 2” of the Discussion as follows:

“Most of the IC1 LOM+IC2 LOM patients show hypomethylation of additional DMRs and have been classified as MLID cases (Table 1). Although further studies are needed to clarify genotype-epigenotype correlations, the current idea to explain the clinical outcome of the MLID patients is the epidominance hypothesis that is based on the mosaic form of the multiple methylation changes in BWS and SRS (Azzi et al., 2009). According to this hypothesis, the main clinical presentation of the patient is caused by the imprinted locus that is mostly affected, while the other affected loci may possibly contribute to atypical features. For example, LOM of GNAS locus has been found associated to 11p15.5 IC2 LOM in MLID patients with BWS and pseudohypoparathyroidism 1B (Sano et al., 2016) or hypocalcemia (Sparago et al., 2019). In SRS-MLID the most affected DMRs other than IC1 are MEST:alt-TSS-DMR and GRB10:alt-TSS-DMR   [12] (Table 1), although we did not find abnormal methylation at these loci in our patient.”